# Robust detection of oncometabolic aberrations by $^1$H–$^{13}$C heteronuclear single quantum correlation in intact biological specimens

Yasaman Barekatain[1], Victoria C. Yan[1], Kenisha Arthur[1], Jeffrey J. Ackroyd[1], Sunada Khadka[1], John De Groot[2], Jason T. Huse[3] & Florian L. Muller [1]✉

Magnetic resonance (MR) spectroscopy has potential to non-invasively detect metabolites of diagnostic significance for precision oncology. Yet, many metabolites have similar chemical shifts, yielding highly convoluted $^1$H spectra of intact biological material and limiting diagnostic utility. Here, we show that hydrogen–carbon heteronuclear single quantum correlation ($^1$H–$^{13}$C HSQC) offers dramatic improvements in sensitivity compared to one-dimensional (1D) $^{13}$C NMR and significant signal deconvolution compared to 1D $^1$H spectra in intact biological settings. Using a standard NMR spectroscope with a cryoprobe but without specialized signal enhancing features such as magic angle spinning, metabolite extractions or $^{13}$C-isotopic enrichment, we obtain well-resolved 2D $^1$H–$^{13}$C HSQC spectra in live cancer cells, in ex vivo freshly dissected xenografted tumors and resected primary tumors. This method can identify tumors with specific oncometabolite alterations such as *IDH* mutations by 2-hydroxyglutarate and *PGD*-deleted tumors by gluconate. Results suggest potential of $^1$H–$^{13}$C HSQC as a non-invasive diagnostic in precision oncology.

[1] Department of Cancer Systems Imaging, University of Texas MD Anderson Cancer Center, Houston, TX 77054, USA. [2] Department of Neuro-Oncology, University of Texas MD Anderson Cancer Center, Houston, TX 77030, USA. [3] Department of Pathology, University of Texas MD Anderson Cancer Center, Houston, TX 77030, USA. ✉email: aettius@aol.com

Metabolic vulnerabilities are emerging as viable therapeutic targets within the framework of precision oncology[1–3]. Well-studied examples of such aberrations include tumors that have mutations in isocitrate dehydrogenase 1 and 2 (IDH1/2) enzymes. As key drivers of tumorigenesis in diverse cancers, such as glioma, cholangiocarcinoma, and leukemia, IDH1/2 mutations have been the foci of a number of experimental and clinical studies[4]. Both cytoplasmic IDH1 and mitochondrial IDH2 enzymes catalyze the conversion of isocitrate to α-ketoglutarate[5]. Recurrent mutations in the active site of these enzymes result in a neomorphic activity that enables NADPH-dependent reduction of α-ketoglutarate to the oncometabolite R-2-hydroxyglurate (2-HG)[6,7]. Thus, the accumulation of 2-HG inside IDH1/2-mutant tumors can reach millimolar level and is the molecular hallmark of this mutation. In glioma, patients harboring IDH mutations have better prognoses compared to the IDH-wild-type group[8,9], and would be the candidate for targeted therapies, such as IDH1-mutant inhibitor, AGI-120 (NCT03564821).

The demand driving the development of drugs targeting tumor metabolism extends beyond IDH mutations. An emerging metabolic vulnerability-based precision oncology is collateral lethality[10,11]. Homozygous deletion of major tumor suppressor genes can result in the collateral deletion of chromosomal neighboring metabolic housekeeping genes, such as the pentose phosphate shunt enzyme 6-phosphogluconate dehydrogenase (PGD) at the 1p36 locus[10–12]. PGD enzyme catalyzes the oxidative conversion of 6-phosphogluconate (6-PG) in the oxidative arm of the pentose phosphate pathway, and its deletion results in >100-fold increased accumulation of 6-phosphogluconate and its hydrolysis product gluconate[12]. As tumors harboring PGD deletions are exceptionally sensitive to the clinical-stage (NCT03291938) oxidative phosphorylation (OxPhos) inhibitor, IACS-010759[12], quantification of 6-PG levels in tumors can be directly diagnostic of PGD-deletion status and concomitant sensitivity to IACS-010759.

A persistent problem for the full realization of the clinical potential of precision oncology, consists of the timely and convenient identification of the subsets of patients with the appropriate genetic alteration for each specific molecular targeted agent. IDH point mutations and the deletion of PGD result in the accumulation of 2-HG and 6-PG, respectively. They constitute representative examples of different targetable genomic alterations, which result in millimolar accumulation of specific metabolites compared to normal brain or wild-type tumors. In principle, such millimolar accumulation of metabolites should be diagnosable with magnetic resonance spectroscopy (MRS)[13–15]. Attempts have been made to diagnose IDH-mutant status by [1]H-MRS[16–18]. However, in practice, there are two limitations to the implementation of this technique to diagnose IDH-mutant and PGD-deletion status. First, the [1]H spectrum of tumors is highly convoluted due to the overlapping chemical shifts of different molecules and signals from informative metabolites, like 2-HG, can be obscured by highly abundant common metabolites[19,20]. Second, even employing the best water suppression pulse sequences, one cannot detect signals from metabolites that have chemical shifts close to the overwhelming water signal in the middle (4.7 ppm) of the spectrum[21]. Because of these biological and technical limitations, as of 2020, no [1]H-MRS protocol for diagnosis of IDH-mutant tumors is FDA approved, or CMS reimbursable.

Here, we demonstrate that it is possible to overcome the specificity limitations of [1]H-MRS through [1]H–[13]C HSQC[21]. This technique deconvolutes the spectrum into two dimensions and harnesses the differences in both [1]H and [13]C-chemical shifts to resolve the signal-peaks of closely related chemical species. Rather than merely peaks in a 1D spectrum, the identity of specific metabolites is characterized by a pattern or constellation of C–H HSQC peaks in a 2D area. The sensitivity of this technique is dramatically higher than standard [13]C-spectroscopy, as the signal is ultimately generated through [1]H-magnetization as well as ultimately read in the [1]H channel[22]. We demonstrated that a phase-sensitive HSQC pulse sequence with short scan times enables detection of 2-HG and 6-phosphogluconate/gluconate in live cancer cells as well as tumors ex vivo; this way, we can reliably identify tumors/cell lines that carry IDH-point mutations or PGD-homozygous deletions. This was achieved without signal enhancing devices (e.g., spinning), [13]C-isotope enrichment, or biochemical metabolite extraction—all despite the high magnetic in-homogeneities inherent in live cells or tumor chunks. These data suggest immediate utility in the setting of just-in-time diagnosis during tumor resection as well as the potential for use in noninvasive MRS setting in vivo.

## Results

**Convolution of [1]H spectra hinders robust detection of 2-HG.** To provide a yardstick for the usefulness of [1]H–[13]C HSQC in the intact biological setting, we first performed detailed studies with conventional 1D [1]H nuclear magnetic resonance (NMR). We performed proof-of-principal experiments in a standard NMR spectrometer (see methods) with standard 5 mm NMR tubes. To provide a solid point of reference, analysis of the [1]H spectrum of the 2-HG pure chemical standard in 10% $D_2O$ phosphate-buffered saline (PBS) was conducted under the exact experimental conditions subsequently used for biological experiments. The [1]H spectrum of the 2-HG standard showed a quartet at 3.92–3.95 ppm for the H2, a convoluted multiplet at 2.11–2.23 ppm for H4 and H4', and multiplets at 1.71–1.79 ppm and 1.87–1.95 ppm for H3 and H3' (Supplementary Fig. 1c). We then took [1]H scans of live mutant-IDH1 cancer cells (HT-1080, NHA mIDH1, SNU-1079 and COR-L105; for a full description of the cell types and mutations, see Methods). As a negative control, we also took [1]H NMR scans of IDH1-mutant cell lines treated with the IDH1 inhibitor AGI-5198[23], which inhibits the production of 2-HG. We found that 2-HG peaks are indistinct, due to broadening and overlap with peaks from other highly abundant metabolites in normal brain and wild-type IDH tumors (Supplementary Fig. 1d–j). For example, glutamine, glutamate, and GABA have chemical shifts close to H4 and H4' of 2-HG, while N-acetyl-Aspartate (NAA) resonances close to frequencies of H3 and H3' protons of 2-HG. Lactate, myo-inositol and phosphocholine and choline have chemical shifts close to H2 peak of 2-HG[19]. Although there are vague hints that peaks may be attributed to 2-HG upon post-hoc analysis, their small signal-to-noise ratio (SNR), broadness, and low amplitude compared to neighboring peaks, in no way, enable confident discrimination between IDH-mutant and WT tumors without previous genetic assignment. Furthermore, the peak broadening is likely to be worse in the MRS setting in vivo, making the confidence in these peaks for specific identification of 2-HG/IDH-mutant tumors even less appealing.

**[1]H–[13]C HSQC allows reliable detection of 2-HG.** To resolve the issue of spectral overlap, we reasoned that metabolites with similar [1]H NMR shifts may be discriminated by the differences in their [13]C NMR shifts via 2D [1]H–[13]C HSQC. After screening various pulse sequences available in Bruker TopSpin (3.5) at MD Anderson's NMR core, we found that the phase-sensitive HSQCETGPSISP3.2 pulse program[24–26] yields high SNR, sharp peaks in the [13]C axis, and distinguishes between positive phase ($CH_3$ and CH peaks) from negative phase ($CH_2$ peaks)—shown

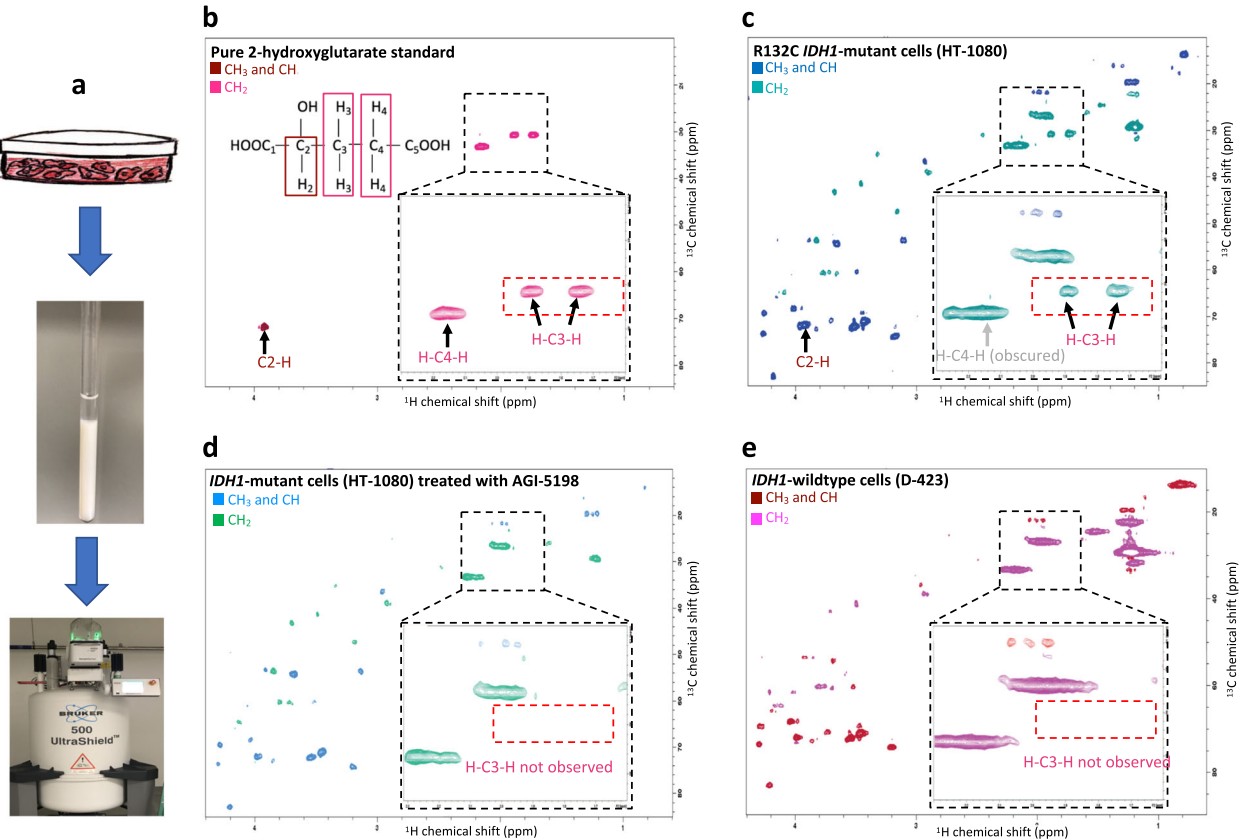

**Fig. 1 The H–C3–H peaks of 2-HG are uniquely detected in live _IDH1_-mutant cells and are eliminated by mutant-IDH1 specific inhibitor treatment.**
**a** Live cancer cells in suspension were placed in an NMR tube for NMR spectral acquisition in 90% PBS and 10% $D_2O$. **b–e** Phase-sensitive $^1H$–$^{13}C$ HSQC spectra using HSQCEDETGPSISP2.3 pulse sequence, where the first color code in each spectrum depicts positive phase peaks ($CH_3$ and CH, e.g., brown in **a**), and the second color code shows negative phase peaks ($CH_2$, e.g., pink in **a**), acquired using 500 MHz Bruker AVANCE III NMR. The black dashed square indicates magnification of the region of the spectrum around H–C3–H peaks of 2-HG, and the red dashed box shows where H–C3–H peaks of 2-HG would be expected in the biological samples (**b–e**). In the $^1H$–$^{13}C$ HSQC spectrum, the protons which are directly bounded with $^{13}C$ atoms are detected ($^1H$-chemical shift in the x-axis, $^{13}C$-chemical shift in y-axis), these carbons and protons are labeled in the 2-HG structure and 2-HG peaks are annotated in the spectrum (**a**). **b** The spectrum of 2-HG chemical standard (7.5 mM) with peak assignments; **c** The $^1H$–$^{13}C$ HSQC spectrum of live R132C _IDH1_-mutant cells (HT-1080). The presence of 2-HG peaks is readily evident in the HSQC spectrum of HT-1080 cells by the H–C3–H peaks. The C2–H peak is very close to the highly abundant myo-inositol peak and barely distinguishable. The H–C4–H peak is obscured by the broad (–CH2–) lipid peak. However, H–C3–H peaks can be easily detected. **d** The spectrum of _IDH1_-mutant cells (HT-1080) treated with 10 μM IDH1 inhibitor (AGI-5198) for 48 h. The H–C3–H peaks of 2-HG disappeared, consistent with the significant reduction in the amount of 2-HG. **e** In contrast to mutant-_IDH1_ cancer cells (HT-1080), the spectrum of live _IDH1_-wild-type cells (D-423) shows a complete absence of H–C3–H peaks of 2-HG.

as different colors (Fig. 1a). Together, these qualities provide an additional layer of confidence in correct metabolite-peak assignments. Using this pulse program, we first characterized the $^1H$–$^{13}C$ HSQC spectrum of the 2-HG chemical standard under the same conditions that we would apply to live cells (Fig. 1a). The 2D $^1H$–$^{13}C$ HSQC spectrum shows resonances for all protons, which directly bound to $^{13}C$ (J1 coupling), with the chemical shift for $^1H$ in the x-axis and $^{13}C$ in the y-axis. Thus, the spectrum completely omits non-H-bonded $^{13}C$ atoms such as carbonyls in carboxylic acids. The H–C2(–OH) of 2-HG resonates at ($^1H$-chemical shift, $^{13}C$-chemical shift) = (3.92 ppm, 72 ppm) and appears with a positive phase (brown in Fig. 1b). Only directly C-bonded H-atom contributes to this signal; thus, the proton on the oxygen remains occulted. The H–C4–H protons of 2-HG resonate at (2.157 ppm, 33 ppm) and appear in the negative phase (pink in Fig. 1b). Finally, the H–C3–H resonances appear as two distinct peaks with chemical shifts of (31 ppm, 1.73 ppm) and (31 ppm, 1.89 ppm) with a negative phase. From this $^1H$–$^{13}C$ HSQC spectrum standard, we also extracted the specific row for each resonance in the 2D spectrum (Supplementary Fig. 2), which provides better visualization of the positive or negative phasing

for each peak which, by extension, indicates the level of sub-stitution at each corresponding carbon.

Having established a reference 2-HG HSQC spectrum, we then employed this pulse sequence to acquire the $^1H$–$^{13}C$ HSQC spectrum of live _IDH1_-mutant cancer cells (HT-1080) resuspended in phosphate-buffered saline with 10% $D_2O$ in a standard 5 mm NMR tube with a 500 MHz NMR. HT-1080 cells harbor the R132C _IDH1_ mutation and overproduce 2-HG[27], which we verified here by mass-spec (Supplementary Fig. 1b). Figure 1b shows the $^1H$–$^{13}C$ HSQC spectrum of live HT-1080 cells acquired over a period of 30 min. In this spectrum, we assigned 2-HG peaks, as learned from the reference standard (Fig. 1a). All 2-HG peaks were clearly visible in the $^1H$–$^{13}C$ HSQC spectrum of live HT-1080 cells. The C2–H peak of 2-HG appears with a positive phase at (3.92 ppm, 72 ppm). This peak of 2-HG has a chemical shift close to the high-abundant metabolite myo-inositol at (3.96 ppm, 72 ppm), which is present in all cancer cell lines examined (Supplementary Fig. 1b). Therefore, it makes differentiating the C2–H peak of 2-HG apart from the C–H peak of myo-inositol hard in in vivo and ex vivo experiments. The negatively phased H–C4–H peaks of 2-HG appears at (2.15 ppm, 33.5 ppm). Finally,

H–C3–H doublet of 2-HG appear at (1.73 ppm, 31 ppm) and (1.89 ppm, 31 ppm) with the negative phase.

Additional to the 2-HG peaks, the spectrum shows peaks for other abundant metabolites such as lactate, myo-inositol, choline, fatty acids, and many others. To make sure we correctly assigned 2-HG peaks in the spectrum, and as the negative control, we scanned live HT-1080 cells treated with 10 μM mutant-IDH1 inhibitor, AGI-5198[23]. The AGI-5198 inhibitor has an IC50 of 160 nM for R132C IDH1-mutant and IC50 of >100 μM for IDH1-wild-type[23]. Therefore, treating IDH1-mutant cells with the nontoxic concentration of AGI-5198 inhibitor results in a significant reduction in 2-HG level, which was confirmed by doing mass-spectroscopy on our HT-1080 cells treated with this inhibitor (Supplementary Fig. 1b). Figure 1c shows the spectrum of live HT-1080 cells treated with 10 μM AGI-5198 for 48 h. Comparing the HT-1080 cells spectrum (Fig. 1b) with the spectrum of HT-1080 cells treated with AGI-5198 (Fig. 1c) revealed that only one pair of peaks was sufficiently distinct from other metabolites to be diagnostically useful. Since the C2–H peak of 2-HG is convoluted with the peak of myo-inositol at (3.96 ppm, 72 ppm), the difference in the HSQC spectrum of HT-1080 cells and HT-1080 cells treated with the inhibitor is indistinguishable at this region of the spectrum. Also, The H–C4–H peak of 2-HG at (2.15 ppm, 33.5 ppm) is obscured by the fatty acid peak at the same location. Only the H–C3–H peaks of 2-HG were not obstructed by peaks from common high-abundance metabolites. The dashed red box in the spectrum shows where we are expecting to see H–C3–H peaks of 2-HG in the spectrum.

To further confirm our observation that the detection of H–C3–H peaks (1.73 ppm, 31 ppm) and (1.89 ppm, 31 ppm) of 2-HG are unique to IDH1-mutant cells, we also acquired the spectrum of live IDH1-wild-type cells (Fig. 1d, Table 1 and Supplementary Fig. 3). Comparing the spectrum of IDH1-wild-type with IDH1-mutant cells, we observed that H–C3–H peaks of 2-HG are absent in the IDH1-wild-type spectra. As we observed in the spectrum of HT-1080 cells treated with mutant-IDH1 inhibitor, C2–H and H–C3–H peaks of 2-HG were convoluted by myo-inositol and fatty acid peaks.

Moreover, we acquired $^1$H–$^{13}$C HSQC spectrum of other IDH1-mutant and -wild-type cells to confirm that we can type IDH1-mutation status of cells by detecting the H–C3–H peaks of 2-HG. Supplementary Fig. 3a, c, and e shows the unique detection of H–C3–H peaks of 2-HG in live NHA mIDH1 cells (mutation in R132H), live SNU-1079, and live COR-L105 (mutation in R132C) cells, while H–C3–H peaks were not detected in spectra of live NHA (control for NHA mIDH1) and live SNU-1079 cells treated with 10 μM AGI-5198 inhibitor under the same experimental conditions. Table 1 shows the list of cells in vitro and ex vivo xenografted tumors that we acquired $^1$H–$^{13}$C HSQC spectra on, the origin of cell lines, their IDH1-mutation status, and whether we detected H–C3–H peaks on their spectrum or not. To the best of our knowledge, this phase-sensitive HSQC pulse sequence has not been applied to biological

**Table 1 The summary of all cell lines that were identified with the IDH1 mutation and PGD-deletion status by $^1$H–$^{13}$C HSQC technique.**

| Cell line | Cell type | IDH1 status | H-C3-H 2-HG | PGD status | Gluconate peak constellation | Biological replicates cell lines in vitro | Biological replicates ex vivo xenografted tumors |
|---|---|---|---|---|---|---|---|
| HT-1080 | Fibrosarcoma | R132C | Yes | WT | No | 7 | 8 |
| SNU-1079 | Cholangiocarcinoma | R132C | Yes | WT | No | 1 | – |
| RBE | Cholangiocarcinoma | R132S | Yes | WT | No | 2 | – |
| COR-L105 | Adenocarcinoma | R132C | Yes | WT | No | 1 | – |
| NHA mIDH1 | Normal Human astrocytes overexpressing mIDH1 | R132H | Yes | WT | No | 2 | – |
| HT-1080 + mIDH1 inhibitor | Fibrosarcoma | R132C | No | WT | No | 2 | – |
| SNU-1079 + mIDH1 inhibitor | Cholangiocarcinoma | R132C | No | WT | No | 2 | – |
| RBE + mIDH1 inhibitor | Cholangiocarcinoma | R132S | No | WT | No | 2 | – |
| NHA mIDH1 + mIDH1 inhibitor | Normal Human astrocytes overexpressing mIDH1 | R132H | No | WT | No | 1 | – |
| NHA | Normal Human astrocytes | WT | No | WT | No | 2 | – |
| D-423 | Glioma | WT | No | WT | No | 5 | 8 |
| LN-319 | Glioma | WT | No | WT | No | 2 | – |
| U-343 | Glioma | WT | No | WT | No | 2 | – |
| G-59 | Glioma | WT | No | WT | No | – | 1 |
| U-87 | Glioma | WT | No | WT | No | – | 2 |
| D-502 | Glioma | WT | No | WT | No | 2 | – |
| NB1 | Neuroblastoma | WT | No | Deleted | YES | 3 | 6 |
| NB1-PGD | Neuroblastoma | WT | No | Rescued | No | 2 | 5 |

We looked at five different IDH1-mutant cell lines (in vitro and ex vivo) that overproduce 2-HG due to this mutation. We were able to detect H–C3–H peaks of 2-HG in all these cell lines (in vitro and ex vivo). Then we treated them with the mutant-IDH1 inhibitor (AGI-5198), which shows significant effects in reducing 2-HG production, and the H–C3–H peaks of 2-HG disappeared after the treatment. The spectrum of live IDH1-wild-type cells shows a complete absence of H–C3–H peaks of 2-HG (in vitro and ex vivo). We also looked at the PGD-deleted cell line (NB1), which significantly accumulates gluconate due to this deletion. We were able to detect gluconate peaks in the spectrum of live cells and ex vivo xenografted tumors. These peaks are absent in the spectra of PGD-rescued and wild-type cells and tumors.

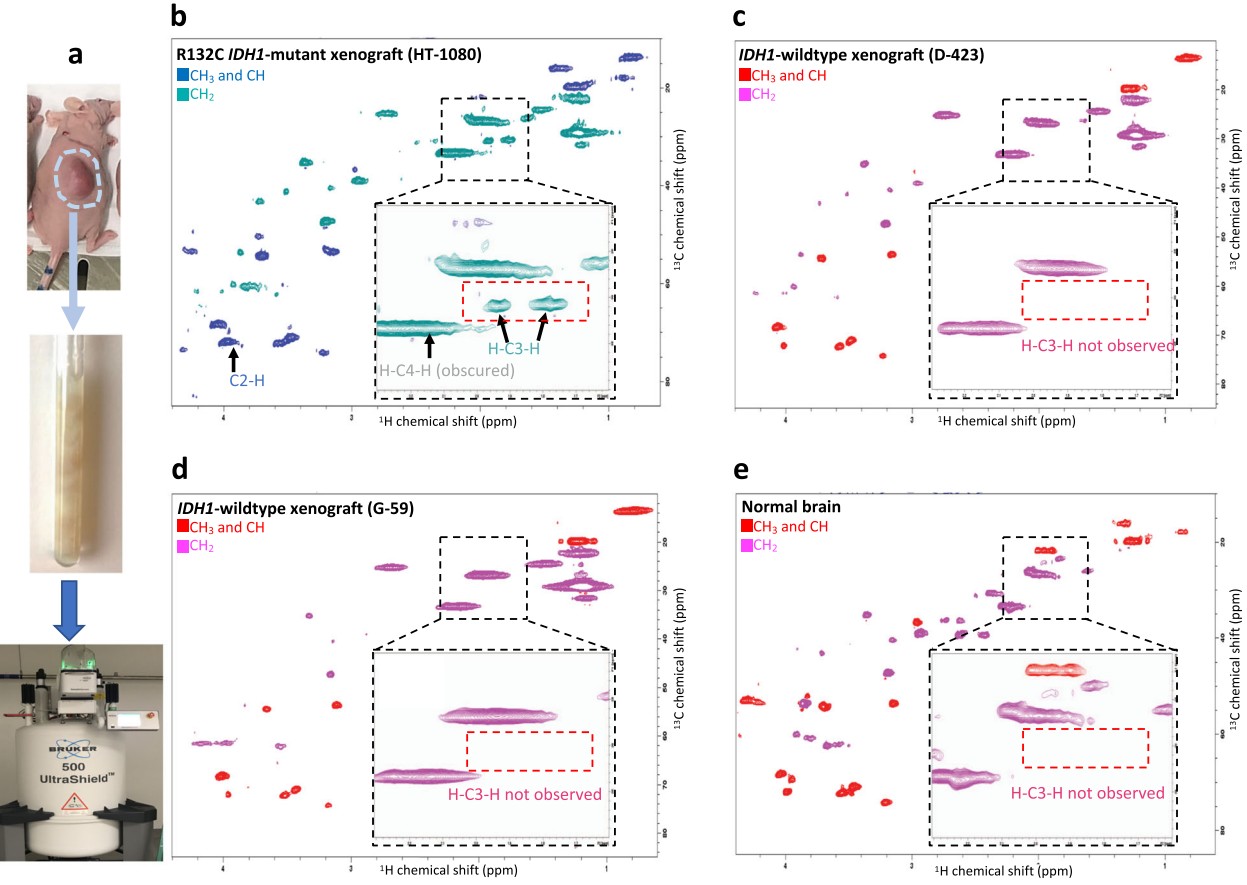

**Fig. 2 The H–C3–H peaks of 2-HG are detected in ex vivo IDH1-mutant but neither *IDH1*-WT xenografted tumors nor normal brain. a** Xenografted tumors were excised and immediately placed in an NMR tube with PBS 10% $D_2O$. The phase-sensitive $^1H$–$^{13}C$ HSQC spectrum was acquired using the HSQCEDETGPSISP2.3 pulse sequence with the same parameters and instrument as used in Fig. 1. **b** The spectrum of ex vivo *IDH1*-mutant xenograft tumor (HT-1080) shows the presence of 2-HG. As in Fig. 1 for live cells, the C2–H peak of 2-HG is very close to the myo-inositol peak, which cannot be distinguished. The broad (–CH2–) lipid peak obscures the H–C4–H peak of 2-HG. However, H–C3–H peaks can be easily detected and are unobscured by other metabolites (pink dashed box). **c–e** As expected, the H–C3–H peaks of 2-HG are absent (pink-dashed box) in the spectra of the ex vivo *IDH1*-wild-type xenograft tumors (D-423 and G-59) and mouse brain.

experiments, and certainly not for diagnosing specific oncometabolites.

We looked at specific row of 2D HSQC spectra to more robustly determined signal to noise ratios and quantitatively compare experimental conditions that alter levels of 2-HG (Supplementary Fig. 3). For the 31 ppm chemical shift, we observed as expected negatively phased doublet associated with H–C3–H peaks of 2-HG in the 2D spectra (Supplementary Fig. 3f). Based on this figure, the specific row of H–C3–H obtained from *IDH1*-mutant cells show negative doublet at 1.74 ppm and 1.89 ppm, the same as the specific row of the 2-HG standard extracted from the same row of the 2-HG HSQC spectrum. These doublets are absent in *IDH1*-mutant cells treated with mutant-IDH1 inhibitor and *IDH1*-wild-type cells.

To further confirm that the detection of H–C3–H peaks of 2-HG is the marker of the *IDH1*-mutation status, we scanned intact *IDH1*-mutant and -wild-type xenografted tumors as well as intact human primary glioblastoma multiforme (GBM) tumors. Figure 2a shows the $^1H$–$^{13}C$ HSQC spectrum of the freshly dissected R132C *IDH1*-mutant xenografted tumor (HT-1080). H–C3–H peaks of 2-HG at (1.78 ppm, 31 ppm) and (1.93 ppm, 31 ppm) are readily detectable in this spectrum. However, these two peaks are absent in the spectra of freshly dissected *IDH1*-wild-type xenografted tumors (Fig. 2b, c). Moreover, H–C3–H

peaks are absent in the spectrum of the intact normal mouse brain (Fig. 2d).

We also looked at the spectra of intact *IDH1*-mutant and -wild-type human GBMs (Fig. 3). We confirmed the *IDH1*-mutation status, as well as high accumulation of 2-HG in the *IDH1*-mutant tumor by mass spectroscopy (Fig. 3b). Then, we looked at the spectrum of the intact *IDH1*-mutant human GBM, where we were able to detect H–C3–H peaks of 2-HG. These two peaks were absent in the spectrum of the intact *IDH1*-wild-type human GBM. Like cells spectra, the C2–H peak of 2-HG in *IDH1*-mutant tumors was convoluted to the C–H peak of myo-inositol, which made detection of C2–H peak of 2-HG in vivo hard and nonspecific. H–C2–H peaks of 2-HG were also obscured in an ex vivo experiment with another metabolite and did not show unique intensity in *IDH1*-mutant tumors. Taken together, these data indicate that $^1H$–$^{13}C$ HSQC detection of the negatively phased doublet peaks of 2-HG at (1.73 ppm, 30.9 ppm) and (1.89 ppm, 30.8 ppm) are a robust biomarker of *IDH1*-mutation status in biological samples.

**HSQC allows screening for diverse oncometabolite aberrations**. Our group has a running interest in passenger-homozygous-deleted metabolic enzymes as points of selective vulnerability for precision oncology[11,12]. Homozygous deletion of metabolic

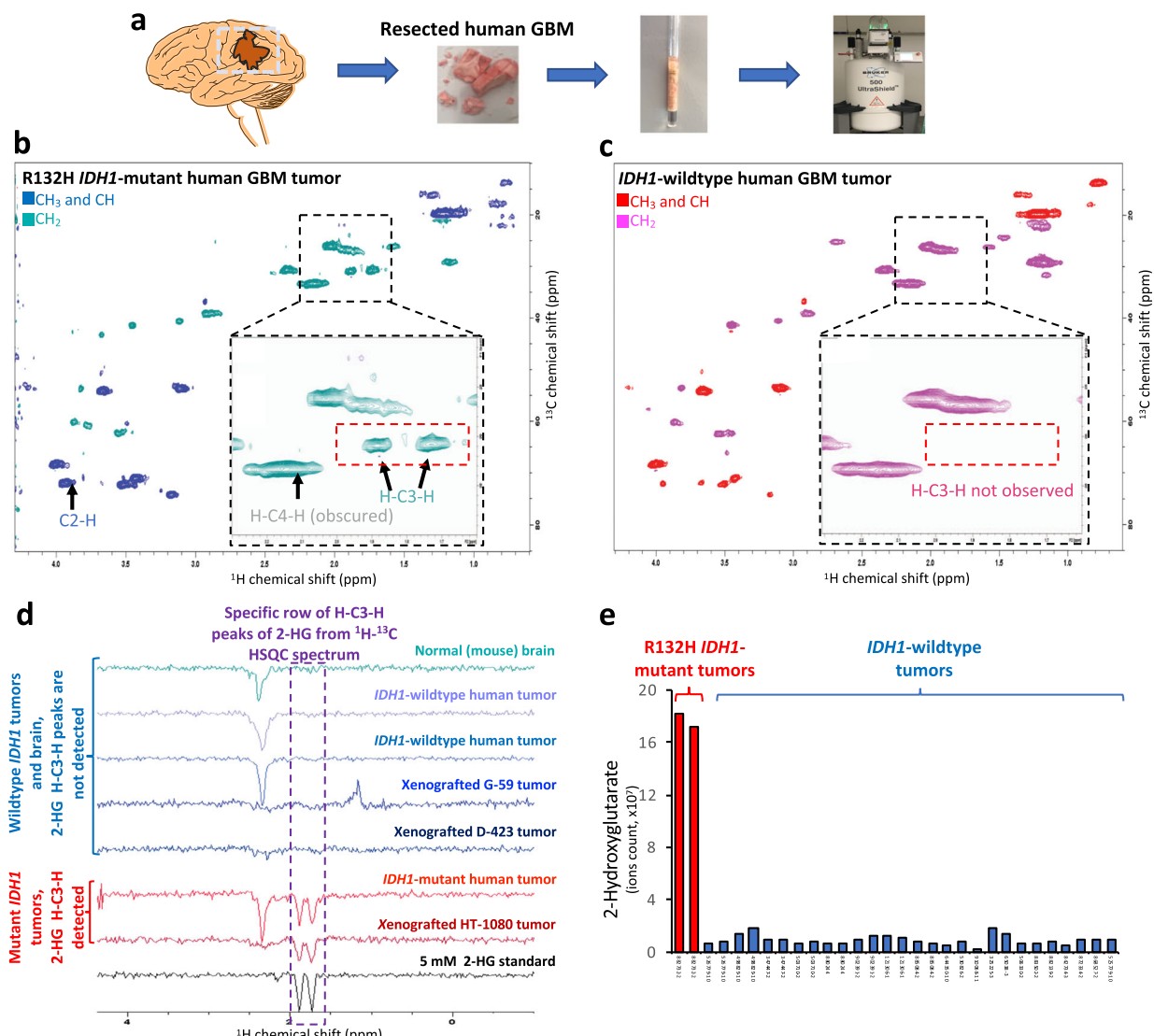

**Fig. 3 The H–C3–H peaks of 2-HG are uniquely detected in *IDH1*-mutant but not *IDH1*-wild-type human GBM ex vivo. a** Flash-frozen GBM tumor resection specimens kept at −80 °C were cut into ~150 mg pieces and placed in a Shigemi NMR tube with 90% PBS and 10% $D_2O$ for NMR spectroscopy studies. **b**, **c** Phase-sensitive $^1H$–$^{13}C$ HSQC spectrum acquired by HSQCEDETGPSISP2.3 pulse sequence, where the first color code in each spectrum shows the peaks, which have positive phase ($CH_3$ and CH) and the second color code shows the peaks which have negative phase ($CH_2$). The black square in the spectrum magnifies the region of the spectrum where H–C3–H peaks of 2-HG are located. The red dashed box shows where we expect to see H–C3–H peaks of 2-HG in the spectrum. **b** The spectrum of an *IDH1*-mutant primary GBM, with the H–C3–H peaks, clearly detectable. The C2–H and H–C4–H are obscured by myo-inositol and (−CH2−) lipid peaks. **c** The spectrum of an *IDH1*-wild-type primary GBM, with no detectable H–C3–H peaks. **d** The specific row of H–C3–H peaks of 2-HG from $^1H$–$^{13}C$ HSQC spectrum of 2-HG standard, mutant-*IDH1* tumors, *IDH1*-wild-type tumors and normal mouse brain. Specific row extracted from the 2D HSQC spectrum of mutant-*IDH1* tumors shows the negatively phased doublet associated with H–C3–H peaks of 2-HG. These two peaks are absent in all *IDH1*-wild-type tumors and normal mouse brain. The spectra are normalized to the noise level. **e** Levels of 2-HG in the *IDH1*-mutant versus *IDH1*-wild-type human GBMs obtained by mass-spec. 2-HG accumulates >20-fold inside *IDH1*-mutant tumors compared to wild-type ones. This figure verifies that H–C3–H are not only uniquely observed in *IDH1*-mutant cells and xenograft tumors, but also in primary GBM tumors.

enzymes stands to cause accumulation of metabolites upstream of the deleted enzymes, and as such, be detectable by NMR-based methods. Having demonstrated that it is possible to genetically type a tumor based on metabolite levels by $^1H$–$^{13}C$ HSQC, we sought to expand its utility to additional oncometabolite aberrations such as those induced by passenger-homozygous deletions of metabolic enzymes. A critical advantage of the $^1H$–$^{13}C$ HSQC spectrum is that the 2D peak patterns for specific metabolites are constellation-like and can serve as the fingerprint identification. The advantage of this compared to $^1H$ is that subtle changes in chemical shifts of specific molecules can happen due to biological microenvironmental changes, such as ionic strength and pH

changes, which confuse the assignment of peaks to a specific molecule. With $^1H$–$^{13}C$ HSQC, the arrangement of peaks even shifted in 1D $^1H$ by microenvironmental effects are unlikely to affect the constellation pattern of peaks in the 2D $^1H$–$^{13}C$ HSQC. This advantage is perfectly illustrated in the detection of elevated gluconate in tumors with *PGD*-homozygous deletion. Figure 4a shows the oxidative pentose phosphate pathway (PPP) where glucose 6-phosphate produced from glycolysis enters into the oxidative arm of PPP and further converts to the 6-PG. *PGD* is the enzyme that is responsible for the oxidation of 6-PG to ribose 5-phosphate. The homozygous deletion of *PGD* results in the accumulation, i.e., 100-fold elevation of 6-PG and its hydrolysis

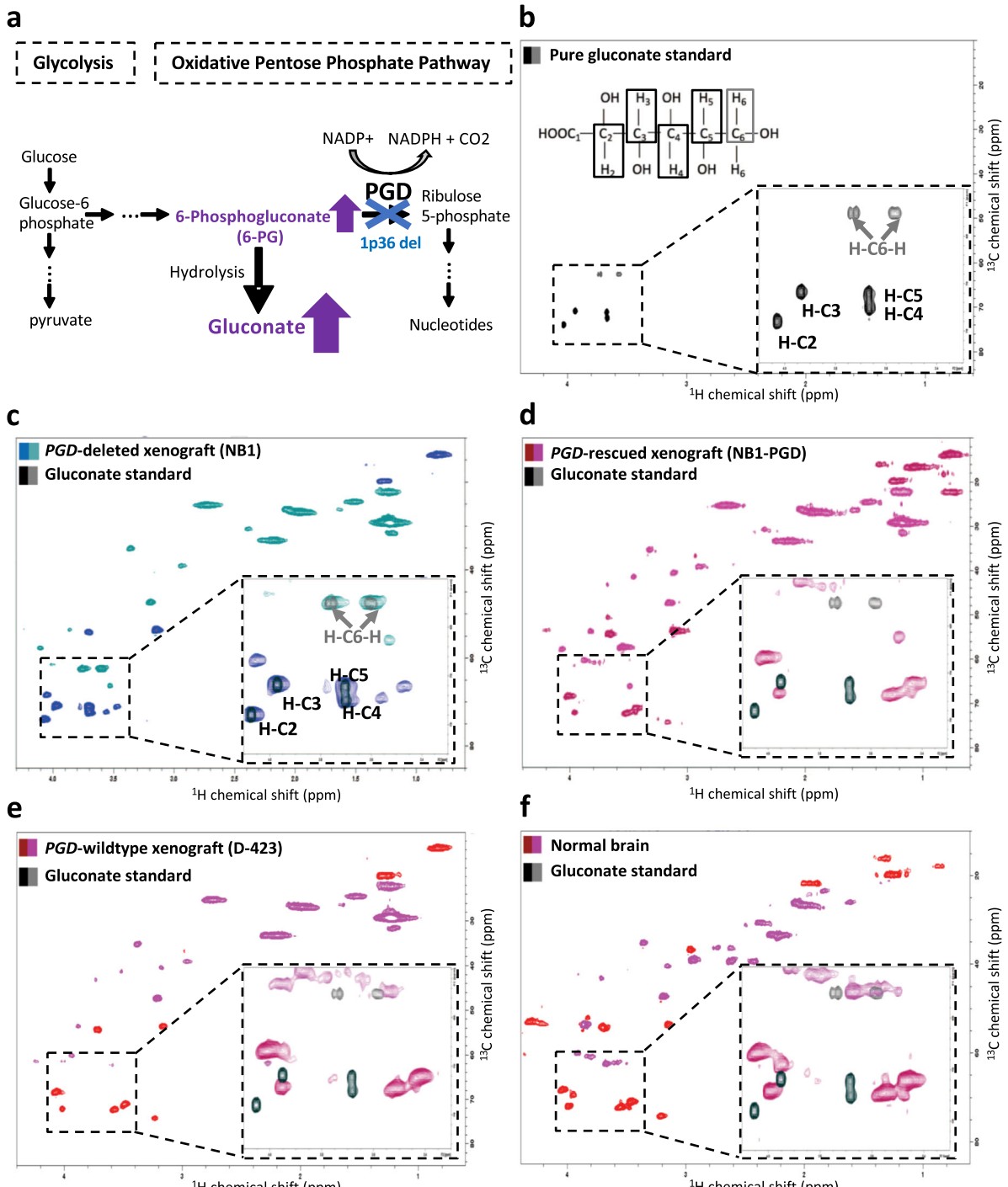

**Fig. 4 Specific HSQC detection of gluconate exclusively in *PGD*-deleted xenografted tumors ex vivo. a** PGD (6-phosphogluconate dehydrogenase) is a key enzyme of the oxidative pentose phosphate shunt, and that can be collaterally deleted in specific cancers as part of the 1p36 tumor suppressor locus[11,12]. High accumulation of the PGD substrate, 6-phosphogluconate, and hydrolysis product, gluconate, have been previously documented by our group in tumors with *PGD*-deletions by mass-spec metabolomics[12]. **b–f** $^1H–^{13}C$ HSQC spectra acquired in a 500 MHz NMR spectrometer using the HSQCEDETGPSISP2.3 pulse program, where the first color code in each spectrum shows the peaks, which have a positive phase ($CH_3$ and CH), and the second color code shows the peaks which have negative phase ($CH_2$). **b** The molecular structure of gluconate and the $^1H–^{13}C$ HSQC spectrum of a gluconate standard at 7 mM concentration in PBS 10% $D_2O$. **c** The $^1H–^{13}C$ HSQC spectrum of a *PGD*-deleted xenografted tumor (NB1) ex vivo. In the black box, the zoomed-in spectrum is overlaid with gluconate peaks from the standard in **a** (black and gray). Alignment of the peak's gluconate standard (black/gray) with the xenografted tumor indicates the successful detection of gluconate. **d–e** $^1H–^{13}C$ HSQC spectra of an ex vivo *PGD*-rescued (NB1-pCMV-PGD) and wild-type (D-423) xenografted tumors. In the black box in the figure, the zoomed-in spectrum is overlaid with gluconate peaks (black/gray) to demonstrate where those peaks are. Although one of the gluconate peaks is obscured with another metabolite, other peaks show unique intensity in *PGD*-deleted tumors. **f** The $^1H–^{13}C$ HSQC spectrum of mouse normal brain. Same as *PGD*-wild-type and rescued tumors, except for the C6-H peak of gluconate, the rest of gluconate peaks were not detected in the spectrum.

product gluconate (Supplementary Fig. 4) compared to *PGD*-intact. Thus, the detection of 6-PG/gluconate can serve as the identification for tumors with this genomic alternation. Although both gluconate and 6-PG are elevated in the *PGD*-deleted cells, the concentration of gluconate is higher than 6-PG, which makes it suitable to detect with MRS. However, the detection of gluconate from the ¹H spectrum is challenging because all protons of gluconate molecules are HO–C–H and H–C–H, which are the common chemical groups in all sugars and other high-abundant metabolites, which results in the convoluted ¹H spectrum. Also, most protons in the HO–C–H group of gluconate resonate very close to water signal at 4.7 ppm which makes the in vivo detection of gluconate difficult. Since ¹H–¹³C HSQC technique integrates the correlation of ¹H and ¹³C atoms, it minimizes the effects of overwhelming water signal (Supplementary Fig. 4). Therefore, peak patterning in the 2D ¹H–¹³C HSQC is ideally suited with the detection of gluconate and typing the *PGD*-deleted tumors. In the Fig. 4b and Supplementary Fig. 5, the ¹H–¹³C HSQC of gluconate standard with specific carbon hydrogen is shown using the phase-sensitive ¹H–¹³C HSQC in which the CH group appears in the positive phase, and the CH₂ group appears in the negative phase. The C2–H peak of gluconate resonates at (4.03 ppm, 74 ppm), the C3–H peak resonates at (3.93 ppm, 71 ppm), the C4–H peak is at (3.66 ppm, 72.5 ppm), the C5–H peak is (3.66 ppm, 71 ppm), and H–C6–H peaks are at (3.55 ppm, 63 ppm) and (3.74 ppm, 63 ppm). Figure 4c, f shows ¹H–¹³C HSQC spectra of ex vivo xenografted tumors and the normal mouse brain. The black dashed box in each spectrum magnifies the region where gluconate peaks are located. To better illustrate where the constellation of gluconate peaks are expected in the 2D spectrum, we overlaid tumor spectra with that of the gluconate chemical standard (black/gray color). Figure 4c shows the spectrum of intact *PGD*-deleted xenografted tumor (NB1), where we observed the near-perfect overlap of the peaks in both dimensions with gluconate standard peaks. In contrast, Fig. 4d shows the spectrum of the ex vivo xenografted *PGD*-rescued tumor (NB1-*PGD*), which was derived from NB1 cells; the *PGD*-deletion was rescued with the atopic expression of *PGD* and effectively generated the NB1 cell line, which is *PGD*-wild-type. All the peaks associated with gluconate are absent in the spectrum of the *PGD*-rescued tumor. Figure 4e, f shows ¹H–¹³C HSQC spectrum of *PGD*-wild-type tumor (D-423) and normal mouse brain where except one peak of H–C6–H doublet, which is obscured by the (–CH2–) metabolites, the rest of C–Hs peaks of gluconate are absent in both spectra. Supplementary Fig. 6 shows spectra of specific rows of the gluconate standard, *PGD*-deleted, -rescued and -wild-type tumors from rows that we expect to observe gluconate peaks. Spectra of extracted rows of *PGD*-deleted tumors show the same peaks as gluconate standard, while these peaks are absent in *PGD*-wild-type tumors and normal brain. Altogether, we show that ¹H–¹³C HSQC is a robust technique to type *PGD*-deletion status of cells and tumors, in addition to *IDH* mutations and elevated 2-HG. This is illustrative of the fact that this technique can type multiple oncometabolic aberrations simultaneously and agnostically.

## Discussion

Cancer metabolism remains an aspirational target for precision oncology[28]. Rapid identification of the subset of patients who are candidates for specific metabolism-based precision oncology drugs remains highly problematic. MRS has the potential to address this challenge, but the utility of this technique (as 1D ¹H-MRS) is currently limited to the detection and quantification of unusually abundant metabolites, which are uninformative for the purposes outlined above. The 1D ¹H-MRS sequence is routinely employed in the clinic; however, it suffers from broadening of peaks and lack of specificity[29]. For example, in the case of 2-HG detection, it is difficult to differentiate chemical shifts of metabolites like 2-HG from similarly structured metabolites such as glutamine and glutamate. Employing spectral editing can improve the convolution/specificity problem, but it results in a sensitivity decrease and loss of global metabolite overview[30]. The ¹³C spectrum has better specificity/spectral resolution compared to the ¹H spectrum; however, it suffers from very low sensitivity due to the low natural abundance of ¹³C (1.1%)[31]. The sensitivity problem can be addressed by using the ¹³C hyperpolarized MRS[32,33]. While this technique is useful for very specific questions, its application is limited to the choice of the hyperpolarized probe, and hence metabolic pathways[21]. Chaumeil et al. used hyperpolarized 1-¹³C α-ketoglutarate to identify *IDH*-mutant tumors noninvasively[34]. However, the technique's sensitivity is limited because of the presence of a 5-¹³C α-ketoglutarate peak, which has a chemical shift close to 1-¹³C 2-HG[34]. Moreover, the use of α-ketoglutarate as the hyperpolarized probe can be restricted due to the hydrophilic nature of this molecule, which can limit its cell membrane permeability[35]. Salamanca-Cardona et al. also used the hyperpolarized 1-¹³C glutamine to identify the *IDH*-mutant tumors[36]. However, the transition of this technique to the clinic has not been forthcoming because of the short T1 half-life of 1-¹³C glutamine[36] and slow glutamine uptake by cells[31]. Correlation spectroscopy (COSY) has been employed in vivo setting to detect 2-HG[30,37]. While this method is capable of detecting 2-HG, its application does not extend to detecting metabolites such as gluconate, which has chemical shifts close to the 4.7 ppm water signal.

Having noted the limitations with other spectroscopic methods, in this paper, we present data to support the case for 2D ¹H–¹³C HSQC in fulfilling the promises of MRS for precision oncology typing and metabolic profiling in vivo. Specifically, we present proof-of-feasibility of metabolite profiling by 2D HSQC NMR in an intact biological setting ex vivo. Compared to the 1D ¹H spectrum of cells, the 2D ¹H–¹³C HSQC spectrum is more resolved, facilitating the assignment of specific peaks to specific metabolites since each 2D HSQC peak is defined by chemical shift information in both ¹H and ¹³C axes. For example, in 2D ¹H–¹³C HSQC spectra of tumors, one can easily differentiate the lactate peak from the broad mobile lipid peak at (1.2 ppm, 30 ppm), which is elevated in high grade GBMs (necrosis)[29], while these two peaks cannot be differentiated in 1D ¹H spectrum. We pioneer a phase-sensitive HSQC (HSQCETGPSISP3.2) pulse sequence to detect specific oncometabolite aberrations in biologically intact oncology samples for the purposes of precision oncology. The sequence not only yields a well-resolved 2D ¹H–¹³C HSQC spectrum but also distinguishes CH₂ from CH₃/CH, providing an additional layer of information. For proof-of-principle, we applied the technique to type tumors that are *IDH1*-mutant and characterized by detection of elevated 2-HG; to demonstrate the wider applicability to metabolic precision oncology, we further show that HSQC can readily identify tumors, which are *PGD*-deleted, by detection of elevated 6-PG and gluconate. While here, we have demonstrated the utility of this technique for two specific metabolic aberrations caused by specific genomic alterations, this technique is applicable to typing of any other metabolites whether genetic or epigenetic in nature. For example, the amino acid glycine is elevated in poor prognosis cases of *IDH1*-mutant glioma patients[38]. We show that glycine is easily visualizable in the ¹H–¹³C HSQC spectra of intact human GBMs in a manner that closely corresponds to the glycine levels as determined in extracts by mass spectroscopy (Supplementary Fig. 7).

While we see the present work as a stepping stone to the application of the phase-sensitive $^1H$–$^{13}C$ HSQC sequence for precision oncology typing by MRS—and our data provide strong proof-of-principal for intact samples ex vivo—legitimate questions may be raised as to whether $^1H$–$^{13}C$ HSQC could be carried out in a full in vivo setting. We reply to this concern by noting that, in fact, in vivo 2D $^1H$–$^{13}C$ MRS method has already been pioneered by other groups[21,39–43], though our study is the first to illustrate its utility for oncometabolite typing. Kato et al. detected $^{13}C$ labeled temozolomide administrated in µM level in a mouse model[42], Chen et al. detected glutamine and glutamate after administrating $^{13}C$ labeled glucose[41], and the de Graaf lab characterized triglycerides in human adipose tissue using this technique[21]. We are the first to demonstrate successful scans with the phased HSQC sequence as well as the successful oncometabolic typing of *IDH1* mutation in intact tumors by HSQC. While throughout this work, we have only relied on natural abundance $^{13}C$, we leave open the possibility of enhancing sensitivity further through the use of specific $^{13}C$ enriched substrates such as glucose and glutamine.

Besides its potential for *IDH* mutation typing in vivo, the HSQC sequence developed here for ex vivo samples could be applied essentially immediately in the (neuro)surgery setting to identify patients with *IDH*-mutant tumors as well as delineating tumor versus normal brain tissue and in the future to type for other potentially targetable metabolic vulnerabilities as they are discovered. Because of the short NMR scan time and no requirement for sample preparation such as chemical extraction, this method would be well suited for the fast just-in-time surgery setting. The detailed metabolite information could complement other imaging and genetic tests such as in vivo MRS as well as genomic sequencing and IHC. The cellular structure of tumors is preserved in this method, which can be used for further genomic and histopathological analysis. Compared to the NMR spectrum of polarly extracted tumor solution, which only detects polar metabolites, the spectrum of the intact tumor has information about polar and apolar metabolites. Indeed, $^1H$–$^{13}C$ HSQC has been employed to compliment $^1H$ metabolic profiling on tumor extracts using very high magnetic field and large amount of samples or on intact tumors using high resolution magic angle spinning[44,45]. We emphasize that the work here was performed in a standard NMR without signal enhancing features such as magic angle spinning and can be implemented in even nonspecialist labs. We do note, however, that a cryoprobe was necessary for optimal sensitivity and that not all NMR facilities would have access to such a device. In sum, $^1H$–$^{13}C$ HSQC stands to dramatically expand our knowledge of tumor metabolism in vivo and to help realize the potential of metabolic precision oncology.

## Methods

**Cell culture**. Cells were cultured at 37 °C in 5% $CO_2$ in DMEM which contains 4.5 g/l of glucose and supplied by 10% fetal bovine serum (Gibco/Life Technologies #16140-071) and 1% Pen Strep (Gibco/Life Technologies#15140-122) and 0.1% Amphotericin B (Gibco/Life Technologies#15290-018). The following established *IDH1*-mutant cancer cell lines were used in this study; cell lines are specified by ExPasy cell-line ID numbers from Cellosaurus, mutation, followed by the cancer they originated. The cell lines used in this study were: HT-1080 (CVCL_0317, *IDH1* R132C, fibrosarcoma), which express *IDH1* at endogenous physiological levels and was kindly provided by Dr. Seth Gammon (Cancer Systems Imaging), SNU-1079 (CVCL_5008, *IDH1* R132C, intrahepatic cholangiocarcinoma) and RBE (CVCL_4896, *IDH1* R132S, intrahepatic cholangiocarcinoma) were kindly provided by Dr. Lawrence Kwang (Translational Molecular Pathology), and COR-L105 (CVCL_1138, *IDH1* R132C, lung adenocarcinoma) was purchased from Sigma (Catalog # 92031918). We also used engineered immortalized normal human astrocytes, NHA m*IDH1* (*IDH1* R132H mutant), which overexpress mutant-*IDH1* and was kindly provided by Dr. Seth Gammon. We also used the following *IDH1* wild-type cell lines: D-423 (CVCL_1160, GBM) and D-502 (CVCL_1162, GBM) kindly provided by Darrel Bigner as we described in more details previously[46], LN-319 (CVCL_3958, Astrocytoma) and U-343 (CVCL_S471,

GBM) from the Department of Genomic Medicine/IACS Cell Bank, MDACC, G-59 (CVCL_N729, Astrocytoma) kindly provided by Dr. K Lamszus, U-87 (CVCL_0022, GBM) obtained from ATCC. Additionally, we used immortalized normal human astrocytes (NHA), which express the wild-type *IDH1* gene and was kindly provided by Dr. Seth Gammon. For the *PGD*-deletion/6-PG study, we used NB1 (CVCL_1440, neuroblastoma) cells (*PGD*-deleted) and NB1-*PGD* cells (NB1 cells with overexpressed *PGD*) from IACS Cell Bank, MDACC. Cell lines were authenticated at MD Anderson Characterized Cell Line Core (CCLC). The Promega 16 high sensitivity short tandem repeat (STR) kit (Catalog number: DC2100) is used for DNA fingerprinting. The STR profiles were compared against online databases and CCLC profiles (more than 5000 profiles). Cell lines were tested negative for mycoplasma by ELISA using the mycoAlert PLUS detection kit (Lonza, Basel, Switzerland). And, every 6 months were tested for mycoplasma by the mycoalert system.

**Preparation of NMR standards**. For assigning metabolite's peaks in the spectrum, we prepared the 7.5 mM L-α-Hydroxyglutaric acid (Sigma–Aldrich, 90790) sample in 90% PBS (Dulbecco's phosphate-buffered saline, 21-031-CV) and 10% $D_2O$ + 3% TSP (Deuterium Oxide 99.9 atom % D, contains 0.75 wt %3-(trimethylsilyl) propionic -2,2,3,3-$d_4$ acid, sodium salt, Sigma–Aldrich). To choose the best $^1H$–$^{13}C$ HSQC pulse program for our study, we prepared the 2-HG sample with 1 mM concentration in 90% PBS and 10% $D_2O$. This low millimolar sample was chosen to evaluate which pulse program is capable of detecting metabolites with low concentration.

For the *PGD* study, standard samples of 6-phosphogluconic acid trisodium salt (Sigma–Aldrich, P7877) and D-gluconic acid sodium salt (Sigma–Aldrich, G9005) were prepared at 7 mM concentration in 90% PBS and 10% $D_2O$. The standard sample concentration was chosen to be in the millimolar level since it is the concentration we had expected in tumors and cells. The final pH of all prepared samples was ~7.

**Sample preparation for in vitro NMR analysis**. Cells were cultured in 15 cm plates. At 90% confluency, cells were harvested and washed once with 1× PBS to remove residual media. Cell number was measured using Trypan Blue (Corning, 25-900-Cl). A suspension of 170 µL cells (40–50 million cells) was then resuspended in 500 µL of a solution containing 90% PBS with 10% $D_2O$ with 3% TSP (Sigma–Aldrich, 450510) and transferred to 5 mm NMR tubes. NMR tubes were sealed with parafilm. The spectral acquisition began immediately after sample preparation. Once measurements were complete, the cell number was measured again using Trypan Blue to verify cell viability during the duration of the experiment. Cells were then returned to 15 cm culture plates to verify normal growth patterns (Supplementary Fig. 8).

**Sample preparation for ex vivo NMR analysis**. All procedures for animal studies were approved by the MD Anderson Animal Care and Use Committee. Nude *Foxn1* BALB/c female mice (bred in house at M.D. Anderson's Department of Experimental Radiation oncology) aged 2–6 months were subcutaneously injected with 5 millions of cells harboring either *IDH1*-mutant or *PGD*-deletion as well as wild-type cells. When the tumor size became large (>1000 mm³), the mouse carrying the subcutaneous tumor was euthanized. Immediately after tumor removal, fresh tumors were cut and placed in NMR tubes containing 300 µL of 90% PBS with 10% $D_2O$. The spectral acquisition began immediately after sample preparation. We also performed NMR on the snapped-feezed human GBM tumors. Collection of the human GBMs was done under the approved institutional review board (IRB) protocol by the MD Anderson (PA15-0940; PI: De Groot), where the informed written consent from patients were obtained. We cut the human tumors into small pieces and place them into the Shigemi tube (Wilmad Labglass, BMS-005B) with 200uL of 90% PBS and 10% $D_2O$, and we acquired the spectrum on them.

**Spectral acquisition**. All spectra were recorded on a Bruker Avance III HD 500 MHz spectrometer, which is equipped with a cryoprobe broadband observe probe, located at MD Anderson Cancer Center. Measurements were taken at 298 K (25 °C). A $^1H$ NMR measurement (zg30 pulse sequence, flip angle 30°, 1-second relaxation delay, 16 scans) was taken for a duration of 1 min and 17 s immediately before $^1H$–$^{13}C$ HSQC spectral acquisition. 2D NMR measurements were obtained using the HSQCEDETGPSISP2.3 pulse program in the Bruker experiment library (1.5-second relaxation delay, F1 = 2048, F2 = 256, four scans) for a duration of 29 min and 44 s.

The spectra were analyzed using Bruker TopSpin 3.1. For the 1D $^1H$ spectra, we did 1st and 2nd order phase correction followed by automatic baseline correction. 2D $^1H$–$^{13}C$ HSQC spectra were phase-corrected as needed, and baseline adjusted at the F1 and F2 axis using abs1 and abs2 commands. Counter level increments were also adjusted to the 1.1, for a total of 28 levels.

The 1D $^1H$ rows of the $^1H$–$^{13}C$ HSQC spectra were extracted using the *rsr* command and specifying the desired row number. Signal to noise ratio (SNR) was obtained by defining the desired signal region by sigf1 and sigf2 commands on Topspin and then using the SINO command. To find the signal to noise ratio for

negative peaks, we first adjusted the negative phased peaks to positive and then select region of interest using the SINO command.

**Polar metabolomic profiling**. We used mass-spectroscopy technique to confirm high level of 2-HG and 6-PG/gluconate in *IDH1*-mutant and *PGD*-deleted cells and tumors. We used Beth Israel Deaconess Medical center (BIDMC) CORE to profile polar metabolites. To extract polar metabolites from cells, we removed media from 10 cm dishes of cells with 90% confluency, and washed with cold PBS, and removed the PBS completely. We then placed dishes into dry ice and added 4 ml of −80 °C precooled 80% methanol in cells and incubated them for 20 min in −80 °C. Then, we scrapped the cells and took the cells lysate into the precooled tubes and spun down cell's lysate in 4 °C at 17,000 × g for 5 min. We then removed the supernatant from the cell pallets and speedvaced it to get the pallets and sent it to BIDMC.

To extract frozen human and xenografted mouse tumors, we put 50 mg of tumors in precooled Fisher Tube #02-681-291 with Qiagen steel beads. Then, we added 1 ml of −80 °C precooled methanol to each tube. To make homogenous solutions, we shook tubes with Qiagen Tissue Lyser at 28 Hz for 45 s in multiple rounds. We then adjusted the final volume of each sample based on 50 mg tissue/2 ml of 80% methanol and incubated samples at −80 °C for 24 h. After 24 h of incubation at 80 °C, we vortexed samples and spun them down at 17,000 × g at 4 °C for 5 min. We then transferred the supernatant to a new tube and speedvaced it to dryness, and sent the pallets to BIDMC. For independent reproduction, tumors were also polar metabolite-profiled through a fee-for-service by Metabolon Inc (Durham, NC).

**Statistics and reproducibility**. The qualitative data presented in this study are not suitable for parametric statistical analysis. The difference between groups are qualitative e.g., detectable vs nondetectable peaks in *IDH1*-mutant vs wild-type; the situation is somewhat similar to western blots with bands in a knockout being absent versus present in WT samples. All bar graphs and spectra represent raw individual biological replicates rather than means with descriptive statistics. The number of independent biological replicates for each sample/tumor is specified in Table 1.

**Reporting summary**. Further information on research design is available in the Nature Research Reporting Summary linked to this article.

## Data availability

Source data for all figures are included as supplementary Data 1 (Metabolomics) and NMR data are available in the Figshare repository [https://figshare.com/articles/NMR_data_zip/12375203]. NMR data can be analyzed using NMR software such as TopSpin, which can be downloaded for free from Bruker website. All other data are available from the corresponding author upon reasonable request.

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

## Acknowledgements

This work was supported by The University of Texas MD Anderson Cancer Center/ Glioblastoma Moon Shot, the Brockman Medical Research Foundation, the SPORE in Brain Cancer (2P50CA127001) funds, the American Cancer Society Research Scholar Award RSG-15-145-01-CDD, the National Comprehensive Cancer Network – Young Investigator Award YIA170032, the Andrew Sabin Family Foundation Fellows Award, and the US National Institutes of Health (1R21CA226301) to F.L.M. We would like to thank Lisa Norberg and Kristin Alfaro–Munoz for help obtaining frozen GBM specimens. We would like to thank Dr. John Asara for metabolite profiling data, Dr. James Bankson, Dr. Gary Martinez, Dr. Mark D. Pagel, Dr. Robin de Graaf, Dr. Jason Stafford, Dr. Dawid Schellingerhout, Dr. Samuel Einstein, Dr. Seth T Gammon, and Dr. Niki Zacharias Millward for their insights into this project. We will also thank Dr. Kumaralal K. Kaluarachchi, the manager of NMR facility at U.T.M.D. Anderson. We would also like to thank Dr. Anna Pawliczek, Dr. Dimitra Georgiou, Yu-Hsi Lin, and Jorge Delacerda for their editorial and technical assistance.

## Author contributions

Y.B. performed metabolomic sample preparation, NMR studies, and data analysis; Y.B., S.K., and K.A. prepared xenografted mouse tumors; J.J.A. performed western blots; J.D.G. and J.T.H. shared genomic data and primary tumor samples; V.C.Y. edited the manuscript; Y.B. and F.L.M. conceived the study and wrote the manuscript.

## Competing interests

The authors declare no competing interests.
