## [Peer Review File · Communications Biology]

Reviewers' comments:

Reviewer #1 (Remarks to the Author):

This paper proposes to phenotype tumors using MR spectroscopy on intact cells or tissues, with a 'classic' setup (i.e. without using additional technical features such as Magic angle spinning (HR-MAS), or hyperpolarization). However, this work still uses a cryoprobe, that is also not a standard equipment, and certainly not in the clinical setting. Therefore the authors should balance their discussion and be more cautious.

The work is innovative in its application (i.e. to try to phenotype tumors), but is not technically innovative.

The technique seems to work fine to phenotype IDH mutations via 2-HG elevation, and this is worth to be published.

However, we can ask ourselves if more subtle changes could be assessed using such a technique.

A major concern in the study design is the use of 50 million cells in 500 microliters, this is not a cell suspension anymore... what about the viability of the cells in such a dense 'suspension'. This seems to me to be a major limitation that should be mentioned along with suggestions to improve the technique. Overall, this is a relevant work that deserves publication after modification of the discussion.

Reviewer #2 (Remarks to the Author):

Comments:

The manuscript by Yasaman et al. describes a method for the detection of oncometabolites (that can reach millimolar level) in live cells and intact tumors ex-vivo. It is based on 2D 1H-13C heteronuclear single quantum correlation technique that can decrease the overlapping of metabolite signals and eliminate the influence of water signal. The sample preparation process is simple (no requirement of chemical extraction), and there is no need for MAS-probe.

Overall, this is a promising non-invasive diagnostic tool for metabolic precision oncology.

Specific comments:

1) Can the two metabolites of gluconate and 6-PG be distinguished in the two-dimensional spectrum?

The spectrum of gluconate standard should be added in the supporting information. And the 1H-13C HSQC spectrum of tumor also needs to be complemented rather than only its specific rows.

2) The abbreviation of 3-(trimethylsilyl)propionic -2,2,3,3-d4 acid should be TSP. The "Bruker" should be "Bruker" in the first line of Spectral acquisition section on page 21.

3) "To choose the best 1H-13C HSQC pulse program for our study, we prepared the 2-HG sample with 1 mM concentration in 100% D2O." in the Preparation of NMR standards section. Why using 100% D2O instead of 90% PBS and 10% D2O (under the same conditions) for sample preparation. This should be explained.

4) The word "projections" of the sentence "The 1D 1H projections of the 1H-13C HSQC spectra were extracted using the rsr command and specifying the desired row number." In the Spectral acquisition section should be "rows".

5) "After screening various pulse sequences available in Bruker TopSpin (3.5) ... we found that ... sharp peaks in the 13C axis ..." in page 7. What types of sequences were tested specifically? This information needs to be supplemented in supplemental information. Compared with other pulse programs, why HSQCETGPSISP3.2 pulse program yields sharp peaks in the 13C axis?

6) The parameters of HSQCEDTGPSISP2.3 pulse program should be described in more detail. The delay of inept transfer, the delay for multiplicity selection and the delay of sensitivity enhancement building block are some key parameters. Is trimming pulse in inept transfer used?

7) In the "Sample preparation for ex-vivo NMR analysis" section, "tumors were cut into small pieces".

Is there a specific size of the pieces required? How to shim the sample?

8) In page 13, "However, the detection of gluconate from the ^1H spectrum is challenging because all the protons of gluconate molecules are C-H ..." This description is inaccurate because there is a group of C6-H2.

Dear Editor:

We would like to thank reviewers for the effort and attention in reviewing the manuscript. We found the comments helpful, constructive and incisive. Below please see changes we made into manuscript as well as our replies to reviewers.

Changes which made into manuscript:

We modified the title (line # 1) and abstract (line # 24) to comply with 15 words and 150 words requirements of the journal. We also added the Statistics and Reproducibility section in the method (line # 506). We also added the description on mycoplasma testing and authentication and origin of cell lines in the method. The rest of changes to the manuscripts are described in the response to reviewers.

Reviewers' comments:

Reviewer #1 (Remarks to the Author):

1) This paper proposes to phenotype tumors using MR spectroscopy on intact cells or tissues, with a 'classic' setup (i.e. without using additional technical features such as Magic angle spinning (HR-MAS), or hyperpolarization). However, this work still uses a cryoprobe, that is also not a standard equipment, and certainly not in the clinical setting. Therefore the authors should balance their discussion and be more cautious.

=>The reviewer is correct to point out that we have used a 500 MHz with a cryoprobe, which increases SNR 2-3 fold. We modified the discussion and abstract sections to point this out explicitly, and discussion to be perfectly upfront about the technical requirements. While the reviewer's point is well taken, we would also point out that cryoprobes are now typically available in many academic centers. For example, in the Houston Medical Center, there is a 500 MHz NMR with cryoprobe at MD Anderson (the one we used for our study), 800MHz NMR with cryoprobe at Baylor University, 600MHz with a cryoprobe at Rice University and 600MHz NMR at McGovern medical center. We are particularly excited that ultrahigh field NMR spectrometers like the Bruker 900MHz are to beginning to be installed in academic centers; we believe that such a device would increase the metabolites-scope of the ^1H - ^{13}C HSQC sequence even further.

While the necessity of the cryoprobe may damp the enthusiasm of our technique with regards to translating to the MRS setting *in vivo*, it is worth to mentioning that there is Bruker MRI cryoprobe commercially available for small animal *in-vivo* imaging experiments.

2) The work is innovative in its application (i.e. to try to phenotype tumors), but is not technically innovative. The technique seems to work fine to phenotype IDH mutations via 2-HG elevation, and this is worth to be published.

However, we can ask ourselves if more subtle changes could be assessed using such a technique.

=> We are grateful that the reviewer deems our work innovative, even if only in its application rather than the underlying technique.

We have utilized HSQC spectra qualitatively for the identification of metabolite aberrations, whereby extreme levels of a specific metabolite distinguish specific tumors and cell lines.

The reviewer is spot on in his comment that the utility of the quantitative application of the HSQC technique in our manuscript remains unexplored. We agree with the reviewer's suggestion a quantitative approach might indeed reveal subtle changes that are meaningful physiologically. A cursory look on the HSQC scan reveals approximately, ~50-70 peaks in intact human tumors (with a reasonable SNR). In fact, we have observed extensive differences between individual tumors of such quantitative differences but our team lacks the mathematical know-how to express this correctly. We therefore reached out to the group at the University of California San Diego (Dr. Gerwick and Dr. Cottrell) who have developed a machine learning approach to quantifying HSQC spectra of extracts of natural products. This algorithm could be employed to quantify metabolites with a high degree of precision in complex intact biological samples scanned with our HSQC protocol. The approach involves acquiring spectra of diverse standards and use the spectra of such standards for simulation and subtraction processes. We have shared our spectra with this group and continue to do so as we progress. We believe this is a potentially very fertile area of investigation, but would require a dedicated manuscript to fully expound.

Besides detection of 2-HG and gluconate in *IDH1* mutant and *PGD*-deleted tumors which are the focus of the paper, the technique can be also used to identify and qualify other metabolites. For example, it is found by Tiwari *et al* that the amino acid glycine is associated to the poor prognosis in glioma patients even ones that harbor *IDH1* mutation. While, the metabolite has not been yet associated with any genomic alterations, it can be used as the biomarker for prognosis. Our technique can readily detect glycine in the spectra of human GBMs as well as cancerous cells and xenografted mouse tumors. To address the reviewer's comment, we included spectra of human GBMs where we detected glycine in them (**Supplementary Fig. S7**). Besides glycine, there are other metabolites which can be detected by our technique, which their role has not been discovered yet.

We modified and added the following sentences in the manuscript (line #: 364):

While here, we have demonstrated the utility of this technique for two specific metabolic aberrations caused by specific genomic alterations, this technique is applicable to typing of any other metabolites whether genetic or epigenetic in nature. For example, the amino acid glycine is elevated in poor prognosis cases of *IDH1*-mutant glioma patients³⁸. We show that glycine is

easily visualizable in the ^1H - ^{13}C HSQC spectra of intact human GBMs in a manner that closely corresponds to the glycine levels as determined in extracts by mass spectroscopy (supplementary Fig. S7).

Supplementary Figure S7

3) A major concern in the study design is the use of 50 million cells in 500 microliters, this is not a cell suspension anymore... what about the viability of the cells in such a dense 'suspension'. This seems to me to be major limitation that should be mentioned along with suggestions to improve the technique.

=> The question of whether cells remain viable during the NMR experiment is highly pertinent and we have now addressed this experimentally. After acquiring an HSQC spectrum using our standard protocol (30 minutes), we recovered the cells from the NMR tube and checked viability by trypan blue and plating efficiency. Shown below is an example of this experiment. The cells remain viable during the course of the NMR experiment as judge by trypan blue, and while we cannot recover all cells that were put in the NMR tube (41M vs 50M), a good number of cells lost are due to pipetting steps whereby cells stick to tips etc. After recovery from the NMR tube, the cancer cells were placed into media and dishes; these cells attached and proliferated just fine. The short duration evidently allows cells to remain alive; we note that cancer cells are in a similar state (actually, even more highly compacted) when prepared for injections into immunocompromised mice for tumor formation experiments. For logistical reasons, harvested cancer cells may remain in this state for at least 30 minutes before being implanted in mice. To address the reviewer's comment, we added the supplementary Figure S8.

Supplementary Figure S8

4) Overall, this is a relevant work that deserves publication after modification of the discussion.

We deeply appreciate the comment and thank the reviewer for this time.

Reviewer #2 (Remarks to the Author):

Comments:

The manuscript by Yasaman et al. describes a method for the detection of oncometabolites (that can reach millimolar level) in live cells and intact tumors ex-vivo. It based on 2D ^1H - ^{13}C heteronuclear single quantum correlation technique that can decrease the overlapping of metabolite signals and eliminate the influence of water signal. The sample preparation process is simple (no requirement of chemical extraction), and there is no need for MAS-probe. Over all, this is a promising non-invasive diagnostic tool for metabolic precision oncology.

Specific comments:

1) Can the two metabolites of gluconate and 6-PG be distinguished in the two-dimensional spectrum? The spectrum of gluconate standard should be added in the supporting information. And the ^1H - ^{13}C HSQC spectrum of tumor also need to be complemented rather only its specific rows.

=> Yes, gluconate and 6-phosphogluconate (6-PG) can be distinguished in the ^1H - ^{13}C HSQC spectrum. We have modified the supplementary figure S5 to address reviewer's comment. The new figure shows ^1H - ^{13}C HSQC spectra of these two metabolites overlaid in the same spectrum. Two metabolites have unique and different ^1H - ^{13}C HSQC spectra, even though they share similar themes. Below please see the updated supplementary figure S5. In the figure, we also include the molecular structure of 6-PG.

Supplementary Figure S5

For the second comment, we are not sure which tumor the reviewer has in mind. In Figure 3d, we showed the specific row projections of H-C3-H peaks of 2-HG from various ¹H-¹³C HSQC spectra. We understand correctly, the reviewer wishes us to show the full ¹H-¹³C HSQC spectra for each of these projections. The projections are from full HSQC spectra which are shown in the figures 1b, 2b, 3b, 2c, 2d, 3c and 2e. The supplementary figure S7 shows spectra for the specific row of H-C3-H peak of 2-HG in cells. The ¹H-¹³C HSQC spectra are also shown in figures 1b, 1c, S3a, S3c, S3e, 1d, S3d, S3b and 1e. The spectra of specific rows for gluconate peaks are shown in the supplementary figure S6. The ¹H-¹³C HSQC spectra for them are in figures 4b-4e, 3b, 2d 3c, S7d and 2e.

2) The abbreviation of 3-(trimethylsilyl)propionic -2,2,3,3-d₄ acid should be TSP. The “Bruker” should be “Bruker” in the first line of Spectral acquisition section on the page 21.

=> Thanks for noting those mistakes, we corrected them accordingly (line #446 and 484).

3) “To choose the best ¹H-¹³C HSQC pulse program for our study, we prepared the 2-HG sample with 1 mM concentration in 100% D₂O.” in the Preparation of NMR standards section.

Why using 100% D2O instead of 90% PBS and 10% D2O (under the same conditions) for sample preparation. This should be explained.

=> This is the valid point, and we prepared the 1mM 2-HG standard in 90% PBS and 10% D2O and tested again, and we found the same results. Therefore, in the manuscript, we corrected our wording (line # 451).

4) The word “projections” of the sentence “The 1D 1H projections of the 1H-13C HSQC spectra were extracted using the rsr command and specifying the desired row number.” In the Spectral acquisition section should be “rows”.

=> Thanks for pointing this out, we have corrected it accordingly (line # 157, 219, 223-225, 305-307, 499, 500, 587 and 589) .

5) “After screening various pulse sequences available in Bruker TopSpin (3.5) ... we found that ... sharp peaks in the 13C axis ...” in the page 7. What types of sequences were tested specifically? This information needs to be supplemented in supplemental information. Compared with other pulse programs, why HSQCETGPSISP3.2 pulse program yields sharp peaks in the 13C axis?

=> Please see the below table that compares different pulse sequences that we tried on the low concentration 2-HG sample. The hsqcetedgpsisp2.3 pulse sequence, the one that we used in this study, yields high SNR and narrow peak in the ¹³C axis.

Name of HSQC pulse sequence	Is it phase sensitive HSQC	Is both H-C3-H peaks of 2-HG detected?	SNR for the specific row of 2-HG OH-C2-H peak	SNR for the specific column of 2-HG OH-C2-H peak	peak width (HZ) for the specific column of 2-HG OH-C4-H peak
hsqcedetgpsisp2.3	YES	YES	12.66	5.82	79.605
hsqcedetgpsp.3	YES	YES	5.16	4.97	86.784
hsqcdietgpsisp.3	YES	YES	6.22	9.84	88
hsqcedetgpsisp2.4	YES	YES	5.02	5.81	88
hsqcedetgpsisp	YES	YES	6.8	9.88	120
hsqcedetgp	YES	YES	3.95	4.11	143
hsqcedetgpsisp2.2	YES	No (one peak was detected)	6.6	6.61	84.444
hsqcedetgpsisp.2	YES	NO	9.18	8.08	82
hsqcdietgpsisp.2	YES	NO	7.36	11.85	86.298
hsqcdietgpsisp.2	NO	YES	7.35	6.37	80
hsqcetgpsisp2.2	NO	YES	10.87	10.39	81.67
hsqcetgpsisp	NO	YES	7.82	12.73	82
hsqcetgpprisp2.2	NO	YES	7.45	8.96	83
hsqcetgpsp	NO	YES	6.98	5.25	83
hsqcetgpsp.2	NO	YES	4.35	4.64	87.578
hsqcetgpsp.3	NO	YES	5.6	6.9	88
hsqcetgpsisp2	NO	YES	6.98	7.36	88
hsqcetgpprisp2.3	NO	YES	7.5	7.19	90.255
hsqcetgpsisp.2	NO	YES	7.03	8.9	91
hsqcetgp	NO	YES	6.52	6.56	93
hsqcetgpsisp	NO	YES	8.79	7.06	127
hsqcdietgpsisp	NO	YES	7.8	5.91	128
hsqcetgprosp	NO	YES	3.8	6.72	136
hsqcdietgpsisp	NO	No	8.34	5.93	141
hsqcetgpsp.2_bbhd	NO	YES	5.46	5.21	N/A

6) The parameters of HSQCEDETGPSISP2.3 pulse program should be described in more detail. The delay of inept transfer, the delay for multiplicity selection and the delay of sensitivity enhancement building block are some key parameters. Is trimming pulse in inept transfer used?

=> The Bruker-defined default HSQCEDETGPSISP2.3 sequence is the double inept transfer using trim pulses during the inept transfer, and shaped pulses for all 180-degree pulses in the ^{13}C channel. Shaped Gradient is also used in the back-inept. The relaxation parameter is equal to 1.5s. The first inept pulse train transfers the magnetization from ^1H to ^{13}C via $(1/J^{13}\text{C}^1\text{H})= 6.89$ milliseconds. There are two delays for the multiplicity selection, $D21= (1/2 * J^{13}\text{C}^1\text{H})= 0.003448$ seconds where $J(^{13}\text{C}^1\text{H})= 145(\text{Hz})$. And $D24= (1/8J^{13}\text{C}^1\text{H})=0.8$ milliseconds, this delay is set to see all multiplicities. In the sequence to increase the sensitivity, the sign of first gradient (G1) alternates in the subsequent scans in order to double the SNR.

7) In the “Sample preparation for ex-vivo NMR analysis” section, “tumors were cut into small pieces”. Is there a specific size of the pieces required? How to shim the sample?

=> The necessity to cut tumors into chunks derives from the technical challenge of inserting tumor/tissue chunks into a 5 mm NMR tube. Tumors were cut into small enough pieces to fit into the 5 mm NMR tube. 90% PBS and 10% D₂O was used to fill the space between tumor chunks. We used the automatic shimming function of the Bruker 500 MHz NMR.

8) In page 13, “However, the detection of gluconate from the ¹H spectrum is challenging because all the protons of gluconate molecules are C-H ...” This description is inaccurate because there is a group of C6-H₂.

=>The reviewer is right, and we have corrected our argument. Please see below the modified sentence. The revised argument is (line # 276):

However, the detection of gluconate from the ¹H spectrum is challenging because all protons of gluconate molecules are HO-C-H and H-C-H, which are the common chemical groups in all sugars and other high abundant metabolites which results in the convoluted ¹H spectrum. Also, most protons in the HO-C-H group of gluconate resonate very close to water signal in 4.7 ppm which makes the *in-vivo* detection of gluconate difficult. The HSQC because it integrates the correlation of ¹H and ¹³C, it minimizes the effects of overwhelming water signal.

REVIEWERS' COMMENTS:

Reviewer #1 (Remarks to the Author):

All comments have been properly addressed. I recommend publication of this article.

Reviewer #2 (Remarks to the Author):

The manuscript is modified and comments are addressed, though it is not technically novel, the application may be a promising alternative for non-invasive diagnostics.